# Peer review of "Integrating Sensor Technologies with Conversational AI: Enhancing Context-Sensitive Interaction Through Real-Time Data Fusion"

_sensors, 2025, doi:10.3390/s25010249_

Round 1
Reviewer 1 Report
Comments and Suggestions for Authors
In this paper, the author examines how sensor technologies intersect with conversational AI models like ChatGPT, the topic is very interesting and the report is well organized and written, However, I have the following comments to make the presentation and content of the report better.
1- lines 24 and 25 are the same!
2- The report does not motivate the problem being solved in the introduction.
3-The open issues should be also highlighted and presented as points with sufficient discussion and it is better to have a separate subsection since they are the fruit of this report.
4- The author needs to make the literature more strong by adding new work and comparison discussion.
Authors could consider these related references in the report:
1. Wang, X., Wan, Z., Hekmati, A., Zong, M., Alam, S., Zhang, M., & Krishnamachari, B. (2024). The Internet of Things in the Era of Generative AI: Vision and Challenges. IEEE Internet Computing, 28, 57-64.
2. Al-Saedi, A.A., Boeva, V., Casalicchio, E., & Exner, P. (2022). Context-Aware Edge-Based AI Models for Wireless Sensor Networks—An Overview. Sensors (Basel, Switzerland), 22.
Author Response
Comment 1- lines 24 and 25 are the same!
Response 1- This redundant line was removed.
Comment 2 - The report does not motivate the problem being solved in the introduction.
Response 2 - The introduction section was completely rewritten to address this issue (pp. 1, 2)
Comment 3 - The open issues should be also highlighted and presented as points with sufficient discussion and it is better to have a separate subsection since they are the fruit of this report.
Response 3 – Bullet points and subsections have now been added throughout the document to better highlight each of the subsections. (pp. 2, 3, 6)
Comment 4 - The author needs to make the literature more strong by adding new work and comparison discussion.
Response 4 – The literature review has been expanded throughout the document and now contains the recommended references (pp. 2, 5, 9, 10, 12)
Authors could consider these related references in the report:
- Wang, X., Wan, Z., Hekmati, A., Zong, M., Alam, S., Zhang, M., & Krishnamachari, B. (2024). The Internet of Things in the Era of Generative AI: Vision and Challenges. IEEE Internet Computing, 28, 57-64.
- Al-Saedi, A.A., Boeva, V., Casalicchio, E., & Exner, P. (2022). Context-Aware Edge-Based AI Models for Wireless Sensor Networks—An Overview. Sensors (Basel, Switzerland), 22.
Reviewer 2 Report
Comments and Suggestions for Authors
After reading this paper, I have the following issues:
In conclusion, I think this paper should be rejected.
Comments on the Quality of English LanguageGood
Author Response
I'm not sure what the "2" in the keywords means. It should be reflected in the text or abstract. It doesn't seem like a proper keyword.
Response: The number 2 has been removed from the keywords and the rest of the keywords have been revised (p. 1)
Throughout the whole paper, it only explores how sensor technologies (such as environmental sensors, biometric sensors, and IoT devices) intersect with conversational AI models like ChatGPT. There is a lack of originality and the innovation is rather low.
Response: The manuscript has been edited throughout to attempt to incorporate this recommendation (pp. 2, 5, 9, 10, 12)
This paper reads more like a popular science article rather than a SCI paper. It fails to elaborate on the issues and doesn't contribute to promoting the industry.
Response: The manuscript has been edited throughout to attempt to incorporate this recommendation (pp. 2, 5, 9, 10, 12)
The typesetting of the paper is very poor. For example, sections 2, 3 are not aligned with sections 4, 5.
Response: The typesetting has been adjusted to attempt to address this issue.
Reviewer 3 Report
Comments and Suggestions for Authors
This is concept paper describing the integration of sensors in the conversational AI agents. The idea is interesting and I do not see its implementation yet. So as a concept this article is ok.
However, from research perspective, I see that it doesn't provide specific details or concrete examples of how this integration works. Including more specific examples, especially when discussing real-world applications, would strengthen the article's practical relevance.
The article touches upon privacy and security concerns related to sensor data but could expand on ethical implications. For example, it could discuss the potential for bias in AI algorithms using sensor data or the societal impact of widespread sensor deployment.
The article talks about the modified pipeline of the sensors' integration. A diagram or model would have aided in the conceptualization of the idea as well as paving the way for its implementation.
Here is more:
The article suggests the use/inclusion of sensors and sensors data as part of the large language models' conversation capabilities. This will allow taking contextual information into account during the man-machine conversation.
The idea is appealing, not yet explored, and original as I am unaware of sensors' real-time integration with LLMs. Although it misses some technical details. For example, how this can be achieved more concretely.
The idea is new and interesting, feasible and will be in vogue soon.
I think the ethical implications and privacy concerns are something that the author could focus more on in the article.
The references are sufficient, relevant and up to date.
I suggest including a figure or two in the article to make it clearer to the reader.
Author Response
Comment 1. This is concept paper describing the integration of sensors in the conversational AI agents. The idea is interesting and I do not see its implementation yet. So as a concept this article is ok.
Response 1. Thank you for this comment.
Comment 2. However, from research perspective, I see that it doesn't provide specific details or concrete examples of how this integration works. Including more specific examples, especially when discussing real-world applications, would strengthen the article's practical relevance.
Response 2. Additional, specific examples have been added throughout the document (pp. 2, 5, 9, 10, 12)
Comment 3. The article touches upon privacy and security concerns related to sensor data but could expand on ethical implications. For example, it could discuss the potential for bias in AI algorithms using sensor data or the societal impact of widespread sensor deployment.
Response 3. An additional section has been added addressing ethics, privacy and security (p. 12)
Comment 4. The article talks about the modified pipeline of the sensors' integration. A diagram or model would have aided in the conceptualization of the idea as well as paving the way for its implementation.
Response 4. A flow chart has been added to address this issue on page 7
Here is more:
Comment 5. The article suggests the use/inclusion of sensors and sensors data as part of the large language models' conversation capabilities. This will allow taking contextual information into account during the man-machine conversation.
Response 5. Thank you for this comment.
Comment 6. The idea is appealing, not yet explored, and original as I am unaware of sensors' real-time integration with LLMs. Although it misses some technical details. For example, how this can be achieved more concretely.
Response 6. This issue was addressed in a revision on page 10
Comment 7. The idea is new and interesting, feasible and will be in vogue soon.
Response 7. Thank you for this comment.
Comment 8. I think the ethical implications and privacy concerns are something that the author could focus more on in the article.
Response 8. An additional section has been added addressing ethics, privacy and security (p. 12)
Comment 9. The references are sufficient, relevant and up to date.
Response 9. Thank you for this comment.
Comment 10. I suggest including a figure or two in the article to make it clearer to the reader.
Response 10. A flow chart has been added to address this issue on page 7
Round 2
Reviewer 2 Report
Comments and Suggestions for Authors
1、Missing punctuation in the text, e.g., lines 161 and 600